# Synergistic Effect of Dipping in Aloe Vera Gel and Mixing with Chitosan or Calcium Chloride on the Activities of Antioxidant Enzymes and Cold Storage Potential of Peach (*Prunus persica* L.) Fruits

M. S. Aboryia [1], Sherif Fathy El-Gioushy [2,*], Rokayya Sami [3,*], Huda Aljumayi [3], Amal Alyamani [4], A. Almasoudi [5] and Mohamed S. Gawish [1]

1 Pomology Department, Faculty of Agriculture, Damietta University, Damietta 34511, Egypt; modyaboryia@du.edu.eg (M.S.A.); msagawishaa@gmail.com (M.S.G.)
2 Horticulture Department, Faculty of Agriculture, Benha University, Toukh 13736, Egypt
3 Department of Food Science and Nutrition, College of Sciences, Taif University, P.O. Box 11099, Taif 21944, Saudi Arabia; huda.a@tu.edu.sa
4 Department of Biotechnology, Faculty of Sciences, Taif University, P.O. Box 11099, Taif 21944, Saudi Arabia; a.yamani@tu.edu.sa
5 Chemistry Department, Faculty of Science, King Abdulaziz University, P.O. Box 42734, Jeddah 21551, Saudi Arabia; asalmasoudi@kau.edu.sa
* Correspondence: sherif.elgioushy@fagr.bu.edu.eg (S.F.E.-G.); rokayya.d@tu.edu.sa (R.S.)

**Abstract:** Peach is a climacteric fruit characterized by a rapid maturation, high respiration level, weight loss, breakdown of texture, and interior browning. Fast tempering of the fruit and subsequent mold expansion caused a negative impact on the marketing. This study was carried out to estimate the synergistic influence of coating with Aloe vera gel (AVG) at 15% or 30% mixed with chitosan (CH) at 1.5% as a kind of natural polymers or calcium chloride (CaCl$_2$) at 3% on physical and chemical features. We investigated the changes in antioxidant enzymes activities of peach fruits *Prunus persica* (L.) Metghamer Sultany. Fruits were kept at $3 \pm 1$ °C and relative humidity (RH) 85–90% for 36 days during two consecutive seasons (2020 and 2021). Results revealed that applying AVG at 30% blended with CH at 1.5% significantly impacted the storage period of peach fruits, reduced the ion leakage (IL), malondialdehyde (MDA), and lessened weight loss. The differences were significant compared to the other treatments and untreated fruits (control) that exhibited the higher values for IL, MDA, and weight loss in the 36th day. Moreover, fruit quality features such as firmness, total acidity (TA), total soluble solids (TSS), and skin color chroma (c*), hue angle (h) were also maintained. Furthermore, this combination was raised of phenolic content, antioxidant capacity (DPPH), antioxidant enzyme activity such as catalase (CAT), peroxidase (POD), and quench the generation of H$_2$O$_2$ and O$_2\bullet-$. It could be concluded that dipping peach fruits in AVG at 30% blended with CH at 1.5% retained the biological features of peach fruit at considerable levels during cold storing. Thus, this effective mixture can be utilized to prolong the storage and marketing period of peach fruits. Nevertheless, a more in-depth analysis is required for this edible coating to be successfully commercialized in the peach fruit post-harvest industry.

**Keywords:** peach; chitosan; Aloe vera-calcium chloride; coating; cell wall enzymes; shelf-life; cold storage

## 1. Introduction

*Prunus persica* (L.) Metghamer, Sultany peach is considered as one of the late varieties that have been cultivated in Egypt, as its survival in the Egyptian market extends until mid-July. The cultivated area of peaches in Egypt is approximately 24,707 ha, and its production is estimated at about 360,723 tons [1]. Peach is a climacteric fruit that maintains

fast maturation after harvesting, owing to the extreme respiration level and weight loss, fruit texture breakdown, and interior browning. The rapid deterioration of the fruit and the subsequent expansion of mold caused a negative impact on consumption and marketing [2]. Many treatments have been noted to increase the post-harvest time of peach, i.e., heat treatment, UV-B radiation [3], 1-methycyclopropene [4], adjusted atmosphere wrapping [5], and edible coating [6]. Recently, food trades have attracted consideration to implementation of natural additives, as a coating [7], in foodstuffs with extra-quality, safety, and environment sustainable expansion [8]. There is a novel method for preserving food and expanding its storing/shelf life which has the ability to improve the food safety through the use of natural or controlled antimicrobial compounds. This is known as bio-preservation [9]. In post-harvest technology, bio-preservation using aims to increase the post-harvest life of vegetables and fruit through the utilization of plant-based stuffs that have long been utilized in food technology. Recently, these herbal products have been used in fresh vegetables and fruits as bio-preservation. Various natural coating products have been examined for minimizing fruit maturation and post-harvest deterioration [10]. One of the most favorable ways to enhance fruit storing/shelf life after harvest is the natural, biodegradable edible coating [11]. The edible coatings, when implemented on fruit covers, act as blocks to weight-loss and gas exchange, decreasing respiration levels and the ripening procedure as well as preserving the fruit features [12]. Aloe vera gel is one of the favorable bio-preservation that has elevated potential to be utilized for most fresh fruits and vegetables. Aloe vera gel is a plant-based natural coating with an acceptable antimicrobial possibility [13]. Overall, 99% of Aloe vera gel forms a colorless gel that comprises of glucomannans (polysaccharides), amino acids, vitamins, and sterols. The utilization of Aloe Vera Gel (AVG) in the post-harvest of vegetables and fruits has significantly increased recently [14]. Applying an AV-based coating declines the dehydration and maintains quality during post-harvest storing [15]. Chitosan was utilized to keep the quality of several fruits such as pomegranate, guava [16], mandarin [17], and others, which is a cationic polysaccharide [18]. Edible films made from chitosan and Aloe vera gel displayed antimicrobial features against post-harvest fruit pathogens [19]. Chitosan is an edible, biodegradable, and non-toxic substance with great coating features [20]. The blended influence of edible coatings versus the fruit post-harvest infections were subjected to researcher's awareness [21]. Furthermore, [22] found a synergistic antifungal effect when applying combined coating from Aloe vera gel and chitosan. Calcium has a central role in cell wall structure and interactions. It can influence the flexibility and permeability of the cell membrane in addition to affecting key functions in the role of membrane, plant signaling, and water relationships [23,24]. Exogenous application of calcium supports the plant cell wall and protects it from cell wall damaging enzymes [25]. Moreover, [26,27] stated that high doses of calcium reduced ethylene creation in apple slices by impeding the transfer of 1-aminocyclopropane-l-carboxylic acid (ACC) to ethylene. The use of $CaCl_2$ markedly declined the prevalence of chilling injury and prolonged the climacteric maturation by reducing the ethylene making levels and fruit softening during storing compared to control [28].

　　　Hence, the purpose of this investigation is to estimate the influence of Aloe vera gel, chitosan, and calcium chloride blended coating on quality parameters, water loss, firmness, and fruit skin color (hue angle, chroma), as well as antioxidant enzymes activities, ion leakage, malondialdehyde, which quench the generation of $H_2O_2$ and $O_2\bullet-$ and storability of peach fruit during cold storing.

## 2. Materials and Methods

### 2.1. Peach Fruit Preparation and Experimental Design

　　　This investigation was conducted during two seasons of 2020 and 2021 on peach (*Prunus persica* L.) var. Mietghemer sultany from a private orchard in Aga city-Dakhlia governorate, Egypt (30.04° N, 31.25° E). Peach fruits were picked on the second week of July in the commercial ripening stage. The fruit was transported immediately to the

horticultural laboratory—Agricultural Faculty of Damietta University. 450 uniform fruit of a certain weight (almost 78 g), obviously free from physical injury and disease, were chosen. The fruit surface was washed with distilled water three times to remove dust, preventing micro-spoilage and reduce the temperature of the fruit. Finally, the fruit was dried with a cotton cloth to remove any water remaining on the surface of the fruits and left about a half hour at ambient air to complete dehydration. The chosen peach fruits were divided into five groups, each group consisting of 90 fruits with three replicates (30 fruits/replicate), each replicate was divided into two main batches. The first one (15 fruits) was used for physical and chemical measurements, while the second batch was used to estimate weight loss. Peach fruits were kept in carton boxes of one kg at $3 \pm 1$ °C and relative humidity (RH) 85–90% for 36 days to investigate chemical and physical characteristics each 12 days. Fruits were soaked for 3 min into one of the subsequent coating mixture treatments:

Distilled water (Control = T1), Aloe vera gel (AVG) at 15% + chitosan (CH) at 1.5% (T2), AVG at 30% + CH 1.5% (T3), AVG at 15% + CaCl$_2$ at 3% (T4), AVG at 30% + CaCl$_2$ at 3% (T5). In all tested treatments, both used substances were mixture in 1:1 ratio ($v/v$) and 0.1 mL/L of Tween 80 was blended as an emulsified agent. Because previous research used high concentrations of AVG ranging from 50% to 100%, we applied the lower concentrations (15–30%) to reduce economic costs.

### 2.2. Manufacturing Aloe Vera Gel (AVG) Extract

Leaves of AVG were collected with the same physical features from plants grown in a local nursery and dipped in a 1% ($w/v$) chlorine solution for one min. Carefully, the pins along the borders of each leaf were removed before splitting the gel tissue to pieces. The gel pieces of leaves were uniformly homogenized in a mixer to get a gummy gel which was refined to remove the fibrous parts. AVG was diluted with distilled water to prepare the required concentrations of 15% and 30% ($v/v$) and then sterilized at 65 °C/30 min and kept at 4 °C before being utilized in the tested treatments [29].

### 2.3. Making Chitosan (CH) Coating

The suspension of CH at 1.5% concentration ($w/v$) was made by homogenizing 1.5 g of CH (Sigma Aldrich, St. Louis, MO, USA) in 100 mL of 1% acetic acid solution using stirring technique [22].

### 2.4. Preparation of the Tested Coating Mixtures

The used coating mixtures were freshly prepared by mixing AVG extract at either 15% or 30% with CH at 1.5% or CaCl$_2$ solution at 3% ($v/v$) according to Vieira et al. [22]. The thickness of the edible film was in general less than 0.3 mm, according Tharantharn [30].

### 2.5. Physical Features
#### 2.5.1. Fruit Weight Loss %

The tested fruits of each box were weighted by a digital balance before (Wi) and at the end (We) of the cold storage. The weight loss of fruit as percentage was obtained applying the subsequent equation: $[(\mathrm{Wi} - \mathrm{We})/\mathrm{Wi}] \times 100$.

#### 2.5.2. Fruit Firmness (lb\inch$^2$)

The pressure tester model FT327 (3–27 lbs.) was used to determine fruit firmness before and after storing period as lb/inch$^2$.

#### 2.5.3. The Peach Fruit Skin Color

The peach fruit skin color was measured using a Spectrophotometer (CM-3600, KONICA, MINOLTA, Tokyo, Japan). The levels of L*, a*, b*, C*, and h were calculated for each fruit. L* value indicates color lightness, and positive (+) and negative (−) values of a* refer to red and green color, respectively, positive (+) and negative (−) values of b* refer to yellow and blue color, respectively. The (C* values) refer to the saturation of the color. The

(h) indicates the red, yellow, green, and blue color. The measures were done twice with 3 replicates (3 fruit) per each treatment. Hue angle (h) = $\tan^{-1}$ (b*/a*), chroma (C*) = (a*$^2$ + b*$^2$) 1/2.

### 2.6. The Chemical Features

#### 2.6.1. Total Soluble Solids Percentage (TSS%)

It was measured in fruit juice using a hand refractometer (Model MASTER-PM Cat.No2393, ATAGO, Tokyo, Japan) at room temperature, and was represented as a percentage.

#### 2.6.2. Total Acidity Percentage (TA%)

It was detected by titrating 5 mL of juice with 0.1 N NaOH using phenolphthalein as a marker to an end-point of pH 8.1 [31].

#### 2.6.3. Total Phenolic Contents (TPC)

TPC in fruit juice of the examined samples were measured by Folin–Ciocalteu reagent, assay using the method of [32]. For quantitatively measurements, a standard curve of gallic acid (0–200 mg/L) was likely made. TPC were expressed as mg gallic acid equivalent (GAE)/g based on FW.

#### 2.6.4. Antioxidant Enzymes Activities (AEAs)

About 5 g of fruit pulp was mixed with 50 mM phosphate buffer (pH 7.8), 0.2 mM of EDTA, and 2% polyvinyl polypyrrolidone (PVPP). After centrifugation at 30,000$\times$ $g$ for 20 min at 4 °C, the supernatant was used for the determines of antioxidant enzymes as explained by [32]. For measurements of catalase (CAT), 3 mL of the reaction blend (12.5 mM $H_2O_2$, 0.2 mL of enzyme extract, and 0.3 mL of 50 mM (pH 7.0) phosphate buffer) was utilized. One unit of catalase enzyme effect was detected based on a reduction of absorbance at 240 nm per min, where the disintegration of $H_2O_2$ has been followed. For peroxidase (POD) activity, 3 mL of the reaction blend contained phosphate buffer (pH 6.0), enzyme extract, 0.1% guaiacol, and 2% $H_2O_2$. After incubation for 5 min, the optical density of blend was noted at 460 nm, where one unit represents an increase of absorbance per min [33]. Measurement was expressed as Unit g-1 protein according to the total soluble protein in samples by the assay of [34].

#### 2.6.5. Ion Leakage (IL%) and Malondialdehyde (MDA) Accumulation

For measuring ion leakage percentage (IL%), a measurement of 5 g of sample (peach peel) was located in 20 mL of 0.4 M mannitol for 3 h/24 °C, then the electrical conductivity was detected initially (R1). After that, the sample was heated in a water-bath at 100 °C for 30 min to quantify the last leakage after cooling the sample at ambient temperature (R2). The ion leakage percentage (IL%) was calculated by the following formula: IL (%) = (R1/R2) $\times$ 100 [35]. Peach pulp (3 g) was ground and combined with 30 mL of metaphosphoric acid (HPO$_3$, 5%) and 500 μL of butylated hydroxytoluene (2%) in ethanol; next, the mixture was homogenized. 1,1,3,3-Tetra-ethoxy-propane (Sigma-Aldrich, St. Louis, MO, USA) was used to vary the amount of TBARS from 0 to 20 mM relative to 0–1 mM malondialdehyde (MDA) as a calibration standard to evaluate MDA accumulation product for the peach samples. The stoichiometry of MDA was calculated throughout the acid-heating step of the assay [36].

#### 2.6.6. $O_2\bullet-$, $H_2O_2$ Production Rate, and Antioxidant Capacity (DPPH%)

Peach pulp (1 g) was mixed in 3 mL of a KH$_2$PO$_4$ buffer 50 mM (pH 7.8) under refrigeration at 4 °C. The reagent was combined with polyvinylpyrrolidone (PVP 1% $w/v$) and immediately centrifuged at 10,000 rpm at 4 °C/15 min $O_2\bullet-$, making levels determined through the formation of nitrite from $NH_2OH$ in the existence of $O_2$ according to the procedure depicted by [37]. At 530 nm, the optical density has been documented. To measure the creation level of $O_2\bullet-$ from the reaction equation of $NH_2OH$ with $O_2$, a

standard curve with $NO_2$ was utilized. The creation level of $O_2 \bullet -$ has been recognized as mmol $min^{-1}$ $g^{-1}$ FW. For determination of $H_2O_2$, 1 g of peach pulp was added to 6 mL of 100% $(CH_3)_2CO$ and immediately centrifuged at $10,000 \times g$ for 15 min at 4 °C, then 1 mL of the clarified supernatant was combined with 0.1 mL of 5% $Ti(SO4)_2$ and 0.2 mL of a $NH_4OH$ solution. The hydrogen peroxide sample was accelerated, and the residue was reduced by adding 4 mL of 2 M $H_2SO_4$ after centrifugation at $10,000 \times$ g for 20 min, then the absorbance was quantified on a photometer at 415 nm. The $H_2O_2$ content was calculated from a standard curve and the fixation rate was shown as mmol $g^{-1}$ FW [38]. The antioxidant capacity (DPPH) of 3 g pulp tissue was blended to 30 mL methyl alcohol and then centrifuged at $10,000 \times g/15$ min. The subsequent obvious extraction (1 mL) was added to 3 mL of 0.1 mM 2,2-diphenyl-1-picrylhydrazyl (DPPH) that dispersed in methyl alcohol. At RT, the reaction blend was hatched in the dark for 20 min. Samples were measured on a spectrophotometer (CM-3600, Konica Minolta, Tokyo, Japan); the wavelength absorbance has been measured at 517 nm and, accordingly, the DPPH radical scavenging effect has been noted. The DPPH% of the sample has been measured by applying the equation of [39].

### 2.7. Statistical Analysis

The Co-Stat software package, Ver. 6.303 (Monterey, CA, USA), was used. The average data of two growth seasons (2020–2021) for the present study have been statistically examined. Data have been analyzed as a completely randomized design (CRD) with triplicate. The means of all treatments have been compared utilizing Duncan's Multiple Range Test at $p \leq 0.05$.

### 3. Results

### 3.1. Weight Loss%, Firmness (lb/inch$^2$), and Peach Skin Color (Chroma, c*, and Hue Angle, h)

Table 1 shows the influence of coating treatments on changes of weight loss, firmness, and peach skin color (c* and h) as physical attributes of fruit quality during a cold storing period at $3 \pm 1$ °C and R.H. 85–90% for 36 days. All treatments showed a significant preservation of fruit quality compared to untreated fruit. The obtained results indicated that the treatment coating of (AVG 30% + CH 1.5%) caused the lowest value of weight loss (23.82%) and chroma (c* = 33.86), and highest value of firmness (10.61 lb/inch$^2$) and hue angle (h = 62.92) compared to the other treatments during the 36 days, while the untreated fruits (control) treatment represented the greatest value of weight loss (40.34%), chroma (c* = 39.19), and lowest value of firmness (6.1 lb/inch$^2$) and hue angle (h = 49.39).

**Table 1.** The effect of AVG coating mixed with CH or CaCl$_2$ on weight loss%, firmness (lb/inch$^2$), hue angle (h), and chroma (c*) of 'sultany' peaches during cold storage.

| Coating Treatments (F2) | Cold Storage Duration (Days) (F1) | | | |
|---|---|---|---|---|
| | 0 | 12 | 24 | 36 |
| weight loss % | | | | |
| Control | 0.00 | 10.59 ± 0.061 [hi] | 22.2 ± 0.538 [de] | 40.3 ± 3.078 [a] |
| AVG 15% + CH 1.5% | 0.00 | 8.62 ± 1.085 [ij] | 19.9 ± 2.083 [ef] | 30.1 ± 2.085 [b] |
| AVG 30% + CH 1.5% | 0.00 | 3.53 ± 2.213 [kl] | 13.7 ± 1.226 [gh] | 23.8 ± 2.442 [cd] |
| AVG 15% + CaCl$_2$ at 3% | 0.00 | 5.25 ± 0.494 [jk] | 18.4 ± 0.776 [ef] | 27.9 ± 2.026 [bc] |
| AVG 30% + CaCl$_2$ at 3% | 0.00 | 4.56 ± 0.683 [jk] | 16.1 ± 1.496 [fg] | 26.4 ± 1.709 [bcd] |
| p-value | F1 = 1.755 | F2 = 1.963 | F1 × F2 = 4.01 | |
| firmness (lb/inch2) | | | | |
| Control | 11.85 | 7.75 ± 0.304 [g] | 7.01 ± 0.341 [h] | 6.1 ± 0.264 [i] |
| AVG 15% + CH 1.5% | 11.85 | 10.23 ± 0.233 [d] | 9.41 ± 0.088 [ef] | 7.7 ± 0.076 [g] |
| AVG 30% + CH 1.5% | 11.85 | 11.36 ± 0.044 [ab] | 11 ± 0.02 b[c] | 10.6 ± 0.133 [cd] |
| AVG 15% + CaCl$_2$ at 3% | 11.85 | 10.43 ± 0.092 [d] | 9.71 ± 0.148 [e] | 9.18 ± 0.092 [f] |
| AVG 30% + CaCl$_2$ at 3% | 11.85 | 11 ± 0.086 [a] | 10.6 ± 0.175 [cd] | 9.71 ± 0.072 [e] |
| p-value | F1 = 0.191 | F2 = 0.214 | F1 × F2 = 0.439 | |

**Table 1.** *Cont.*

| Coating Treatments (F2) | Cold Storage Duration (Days) (F1) | | | |
|---|---|---|---|---|
| | **0** | **12** | **24** | **36** |
| hue angle (h) | | | | |
| Control | 79.02 | $59.62 \pm 0.291$ [h] | $54.3 \pm 0.003$ [j] | $49.4 \pm 0.881$ [k] |
| AVG 15% + CH 1.5% | 79.02 | $61.62 \pm 0.58$ [fg] | $57.1 \pm 0.000$ [i] | $55.6 \pm 0.003$ [j] |
| AVG 30% + CH 1.5% | 79.02 | $70.65 \pm 0.003$ [b] | $65.1 \pm 0.000$ [d] | $62.9 \pm 0.000$ [ef] |
| AVG 15% + CaCl$_2$ at 3% | 79.02 | $63.75 \pm 0.577$ [ef] | $61.3 \pm 0.437$ [g] | $57.2 \pm 1.151$ [i] |
| AVG 30% + CaCl$_2$ at 3% | 79.02 | $66.92 \pm 1.166$ [c] | $62.5 \pm 0.176$ [efg] | $59.4 \pm 0.577$ [h] |
| *p*-value | F1 = 0.624 | F2 = 0.698 | F1 × F2 = 1.429 | |
| chroma (c*) | | | | |
| Control | 32.49 | $36.84 \pm 0.288$ [def] | $38.7 \pm 0.28$ [c] | $48.7 \pm 0.291$ [a] |
| AVG 15% + CH 1.5% | 32.49 | $33.34 \pm 1.166$ [jk] | $36.6 \pm 0.271$ [efg] | $42.2 \pm 0.285$ [b] |
| AVG 30% +CH 1.5% | 32.49 | $32.27 \pm 0.000$ [ij] | $34.5 \pm 0.294$ [fg] | $35.6 \pm 0.003$ [cd] |
| AVG 15% + CaCl$_2$ at 3% | 32.49 | $34.27 \pm 0.606$ [k] | $36.2 \pm 0.291$ [hi] | $39.9 \pm 0.291$ [gh] |
| AVG 30% + CaCl$_2$ at 3% | 32.49 | $35.02 \pm 0.005$ [hi] | $36.4 \pm 0.005$ [efg] | $37.5 \pm 0.28$ [dc] |
| *p*-value | F1 = 0.448 | F2 = 0.501 | F1 × F2 = 1.027 | |

The standard error ($\pm$SE of *n* = 3) is shown by the vertical bars and the alphabetical notes indicated the significance at $p \leq 0.05$ among treatments in each storing time. The mean of both trial seasons (2020 and 2021) was analyzed by using Duncan's Multiple Range Test.

*3.2. TSS% and TA%*

Figure 1 shows the variation in two chemical properties of 'Sultany' peach (TSS% and TA) affected by AVG coating mixed with CH or CaCl$_2$ during 36-day cold storage period. Apparently, TSS% significantly raised in the control fruit during the storing phase. Nevertheless, the (AVG 30% + CH 1.5%) treatments presented the smallest changes in TSS% compared with untreated fruits (control) and other treated fruit. The chemical properties result for treated fruit compared with the control one demonstrated a steady rise in TSS% throughout the 36 days. However, decreases in TA% during storing duration in fruit handled with (AVG 30% + CH 1.5%) were found. This made the smallest differences in TSS during the cold storing period compared with other treatments and the primary levels at harvest time, with 10.4% TSS. A steady TA level (0.83%) was retained on the 36th day of the storing period compared with the preliminary level (1.17%) at the harvest period.

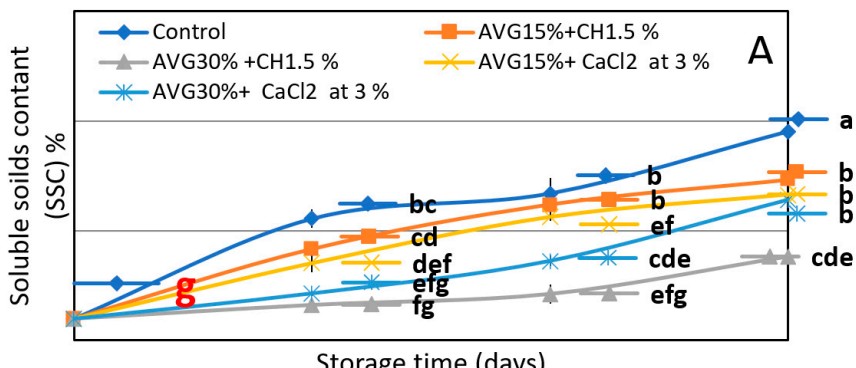

**Figure 1.** *Cont.*

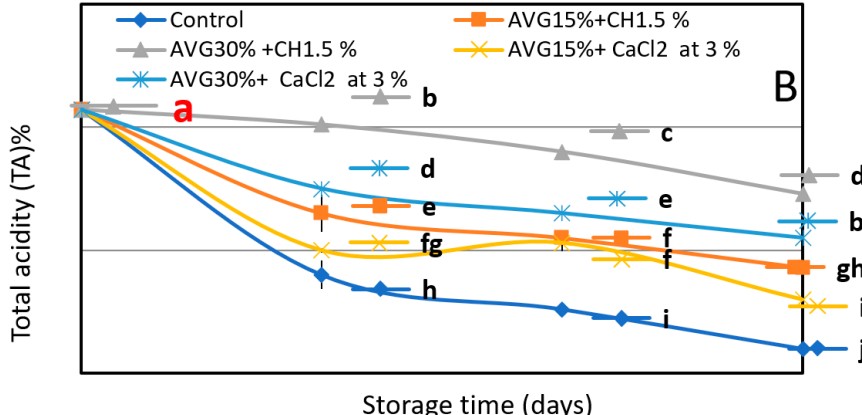

**Figure 1.** The effect of coating with AVG at 15 and 30% mixed with CH at 1.5% or CaCl$_2$ at 3% on TSS% (**A**), and TA% (**B**) of 'sultany' peaches during cold storing at 3 ± 1 °C and 95 ± 1% RH for 36 days. The standard error (±SE of *n* = 3) is shown by the vertical bars, and the alphabetical letters indicated the significance at *p* ≤ 0.05 among treatments in each storing period. The average of both trial seasons (2020 and 2021) was analyzed by using Duncan's Multiple Range Test. *p*-value of TSS%, F1 = 0.448 F2 = 0.501 F1 × F2 = 1.027 and TA%, F1 = 0.033 F2 = 0.037 F1 × F2 = 0.076.

### 3.3. Analysis of Ion Leakage (IL%) and Malondialdehyde (MDA) Accumulation

Figure 2 shows the incremental increase in IL% and MDA accumulation in peach tissue during the storing period (36 day), based on the storing duration and treatments. Peach fruits coated with a combination of (AVG 30% + CH 1.5%) significantly minimized the levels in IL% and MDA (19.23% and 0.28 μM g$^{-1}$ FW) compared with the initial values. The greatest IL and MDA content (48.62% and 0.49 μM g$^{-1}$ FW) was noticed in the raw fruits (control) at the end of the cold storing period. The variations among the treatments also occurred on the 12th day and developed to be more pronounced during the storing period until the ending of the investigation. Thus, these findings showing that the blended influence of both coating reasons (AVG 30% + CH 1.5%) retained the membrane reliability of treated peaches.

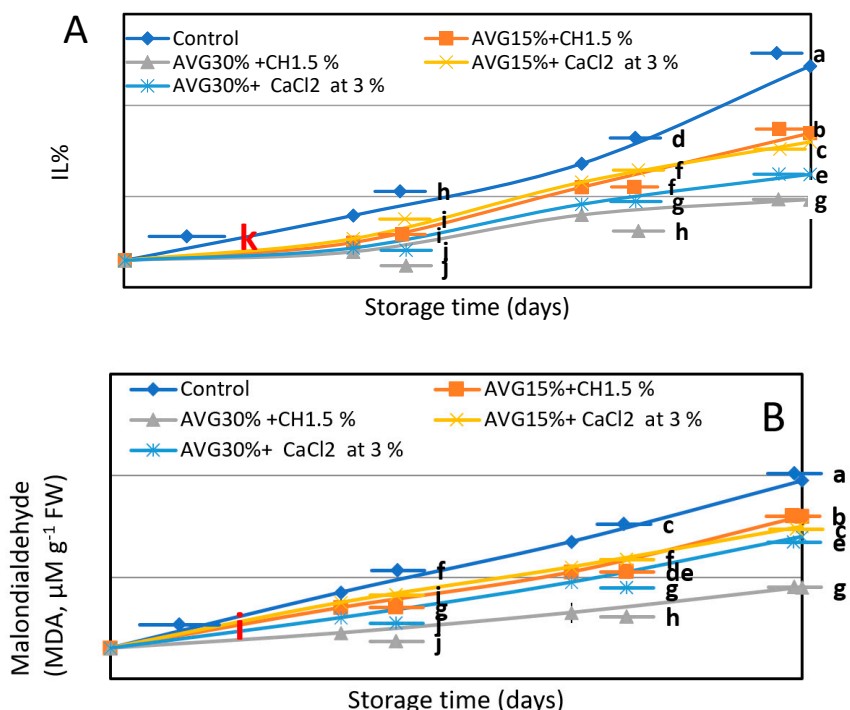

**Figure 2.** The effect of coating with AVG at 15 and 30% mixed with CH at 1.5% or CaCl$_2$ at 3% on IL% (**A**), and MDA% (**B**) of 'sultany' peaches during cold storing at 3 ± 1 °C and 95 ± 1% RH for

36 days. The standard error (±SE of *n* = 3) is shown by the vertical bars and the alphabetical letters indicated the significance at $p \leq 0.05$ among treatments in each storing period. The average of both trial seasons (2020 and 2021) was analyzed by using Duncan's Multiple Range Test. *p*-value of IL%, F1 = 0.506 F2 = 0.566 F1 × F2 = 1.156 and MDA%, F1 = 0.009 F2 = 0.011 F1 × F2 = 0.022.

### 3.4. $H_2O_2$, $O_2$ Production, and DPPH Reduction

Figure 3 shows the differences in production rates of $H_2O_2$ and $O_2 \bullet-$ and antioxidant capacity DPPH% that were influenced by different treatments and the cold storing duration up to 36 days. The obtained results indicate the production rates of $H_2O_2$ and $O_2 \bullet-$ increased clearly in all treatments, except for peach fruits coated with a combination of (AVG 30% + CH 1.5%) that slightly improved the production rates of $H_2O_2$ and $O_2 \bullet-$ beginning from the 12th day until the end of the storing period, while registered the lower production rate of $H_2O_2$ (0.26 mmol min$^{-1}$ g$^{-1}$ FW) and $O_2 \bullet-$ (0.53 mmol min$^{-1}$ g$^{-1}$ FW) until the end of the experiment compared with the other treatments and control. The antioxidant effects of peach fruit influenced by all treatments and storing time (36 days) was recorded as the available radical scavenging activity effect via the DPPH% reduction process. When considering the antioxidant effects, the treated fruits showed different and intense levels of antioxidant synopsis compared to (control), which indicated the lower effect with 33.37% in the 36th day. The higher DPPH% activity was demonstrated by a combination of (AVG 30% + CH 1.5%) treatment, which had a percent inhibition of 39.71% at 36th day. Therefore, (AVG 30% + CH 1.5%) edible coating combination had considerably generated the DPPH radical scavenging effect in peach fruit.

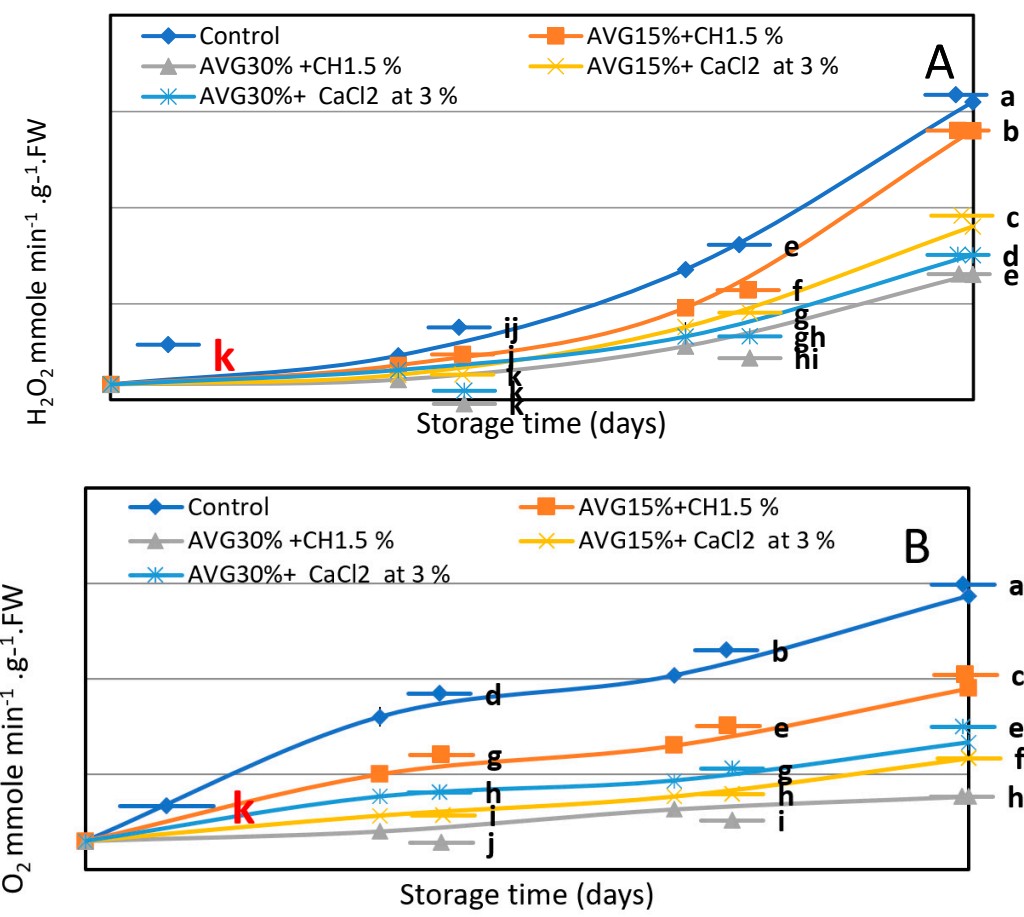

**Figure 3.** *Cont.*

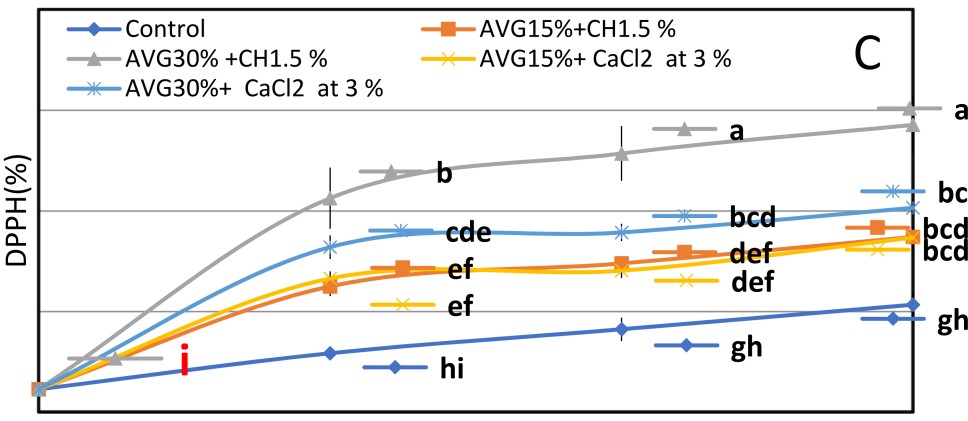

**Figure 3.** The effect of coating with AVG at 15 and 30% mixed with CH at 1.5% or CaCl$_2$ at 3% on H$_2$O$_2$ mmole min$^{-1}$·g$^{-1}$ FW (**A**), O$_2\bullet-$ mmole min$^{-1}$ g$^{-1}$ FW (**B**), and DPPH % (**C**) of 'sultany' peaches during cold storing at $3 \pm 1$ °C and $95 \pm 1$% RH for 36 days. The SE ($\pm$SE of *n* = 3) is shown by the vertical bars and the alphabetical letters indicated the importance at $p \leq 0.05$ among treatments in each storing period. The average of both trial seasons (2020 and 2021) was analyzed by using Duncan's Multiple Range Test. *p*-value of H$_2$O$_2$, F1 = 0.009 F2 = 0.011 F1 $\times$ F2 = 0.020, O$_2\bullet-$, F1 = 0.010 F2 = 0.011 F1 $\times$ F2 = 0.023 and DPPH, F1 = 0.84 F2 = 0.94 F1 $\times$ F2 = 1.93.

### 3.5. Antioxidant Enzyme Activity and TPC

Figure 4 indicates the differences of antioxidant enzyme activities (AEAs) and total phenol content (TPC). AEAs increased gradually from the beginning of the 12th day of cold storing period up to the 36th day. Obviously, the AEAs showed a significant interaction when the combination of (AVG 30% + CH 1.5%) coated fruits treatment and storing period were assessed as a testing factor. Clearly, a combination of (AVG 30% + CH 1.5%) heightened the AEAs in peach. The activity multiplied (CAT, 20.15; POD 27.16 Unit g-1 protein) until the 36th day of testing. Polyphenols are deemed to be essential secondary metabolites that have a strong influence on the quality (e.g., taste, color, and acidity) of fruits and have antimicrobial and antioxidant features. Throughout the whole storing period, TPC was noticed to be various in all treatments and controls. In the case combination of (AVG 30% + CH 1.5%) coated fruits, the TPC was induced throughout the cold storing time of up to 36 days. The increased content of phenols in coated fruits with combination of (AVG 30% + CH 1.5%) could be ascribed to the improved ethylene production level throughout the storing period.

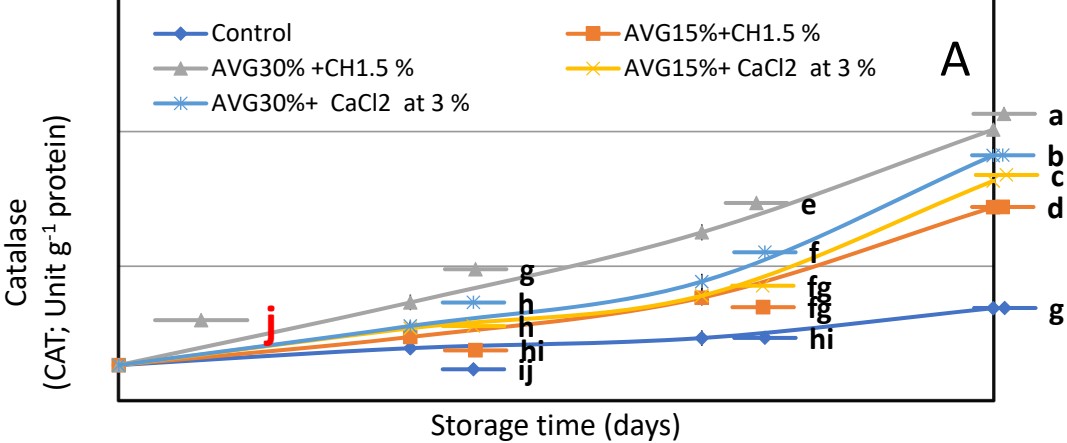

**Figure 4.** *Cont.*

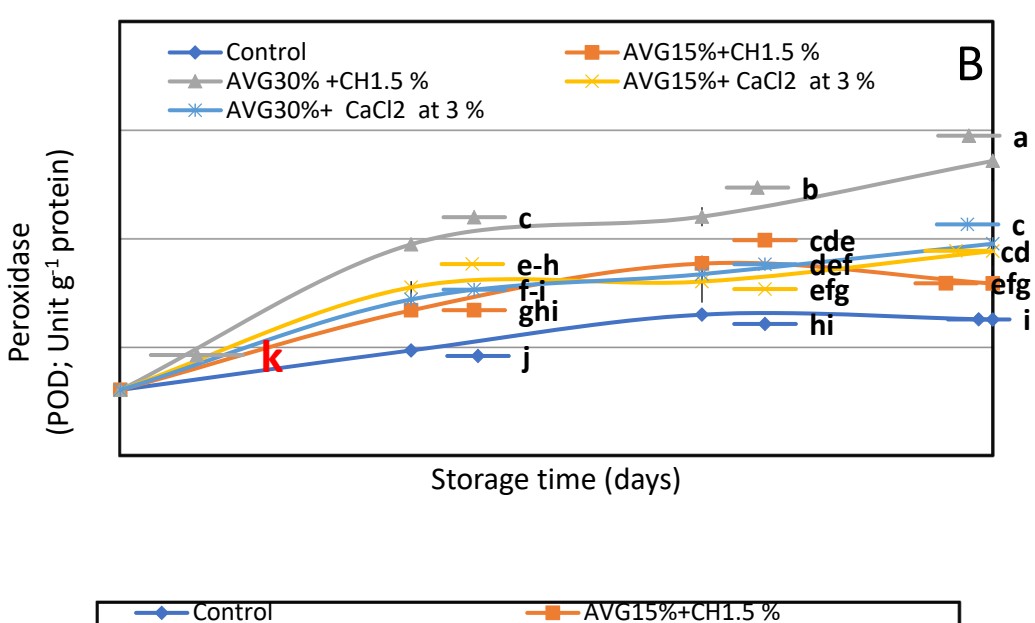

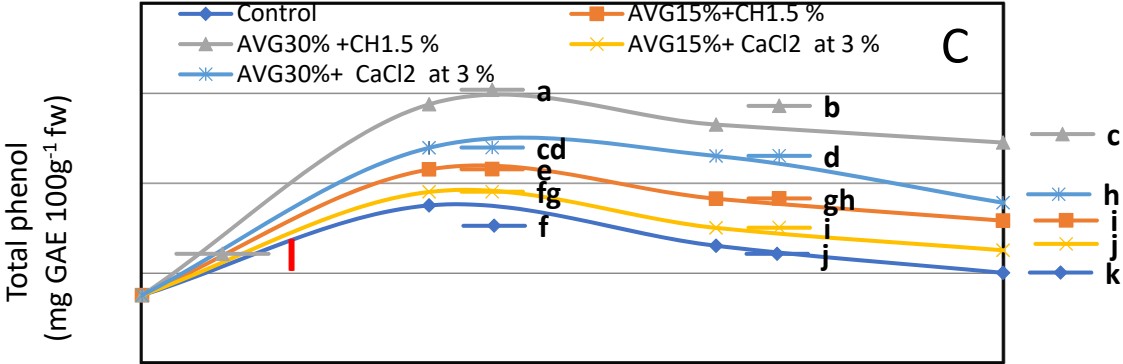

**Figure 4.** The effect of coating with AVG at 15 and 30% mixed with CH at 1.5% or CaCl$_2$ at 3% on CAT (Unit g$^{-1}$ protein) (**A**), POD (Unit g-1 protein) (**B**), and TPC (mg GAE 100 g-1 fw) (**C**) of 'sultany' peaches during cold storing at 3 ± 1 °C and 95 ± 1% RH for 36 days. The SE (±SE of $n$ = 3) is shown by the vertical bars and the alphabetical letters indicated the significance at $p \leq 0.05$ among treatments in each storing period. The average of both trial seasons (2020 and 2021) was analyzed by using Duncan's Multiple Range Test. $p$-value of CAT, F1 = 0.530 F2 = 0.593 F1 × F2 = 1.214, POD, F1 = 1.064 F2= 1.189 F1 × F2 = 2.43 and TPC, F1 = 0.88 F2 = 0.99 F1 × F2 = 2.02.

## 4. Discussion

Current results indicated that reducing weight loss and maintaining firmness and skin color of peach fruit through the 36 days cold storage could greatly be aquired by coating combination of AVG 30% and CH 1.5%. The fruit texture is a necessary quality attribute for fruit marketable and consumer acceptability. The major explanation of weight loss and decreasing firmness is the transpiration and respiration of water from fruit tissues to the surrounding air. It was theorized that AVG coating formed a moisture barrier and minimized transpiration of water, thus delaying weight loss and maintaining firmness [40,41]. It was reported that AVG coating treatment used on the surface of the fruit enhanced the fruit's skin obstruction of gas exchange and reduced respiration levels [42] and ethylene biosynthesis [40,43]. During storing, weight decline and fruit shrinking are the results of weight loss percentage [44]. Nevertheless, it is well reported that edible coating determines weight loss and preserves fruit weight throughout cold storage [45]. AVG, when blended with other edible coatings, improves water preserving capacity [46]. Fungal growth disrupts the integrity of the cuticle, causing increased evaporation and greater weight loss and

softening. This reinforces [22], who recorded weight increase and firmness in blueberry fruits treated with CH that was incorporated into AVG coating. The tempering of fruits is owing to the degradation in the intracellular materials and cell wall formation [47]. It is a biochemical process in which starch and pectin are enzymatically hydrolyzed such as wall hydrolases. Maintaining the firmness of peach coated with CH could be owing to its greater antifungal effects and coverage of the epidermis, decreasing infection, breathing, and other maturation procedures during storing, corresponding to earlier reports on papaya and sweet cherry with CH and AVG coated [20,48].

CH coating treatments delay the product of anthocyanins and pigments in peach fruits [49]. Likewise, the edible AVG coating enhances the fruit pigments by inhibiting enzymatic browning [50]. Similarly, the color changes in sweet cherries that were immersed in AVG and were also delayed [48]. In addition to the synergistic activity on the color retaining, the AVG blended into the CH coating decreases the effects on the respiratory rate, which leads to an interruption in the fruit maturation and thus color changes [50]. These results agree with [51], who noted extra color retention in pumpkin seeds by improving the levels of the CH treatment coating. Similarly, [45] examined that polysaccharide-based combined coatings were efficient in decreasing color deprivation in mango fruits. In the current study, AVG + CH efficiently reduces the respiratory level and the browning reactions, whereby the color of the peach fruit is retained during storage. This investigation indicates the significance of CH and Aloe vera coatings in decreasing weight deficit and preserving the strength and skin color of peach fruit during cold storage.

In this study, TSS improved in all treatments during cold storage. Regardless, the increase was significantly ($p \leq 0.05$) smaller in AVG 30% + CH 1.5% treated fruits. During the storing, starch degradation is altered to glucose, fructose, and sucrose, which are known as the key sugars in fruits [52]. Nevertheless, CH coating minimizes starch degradation and sugar content [53]. Furthermore, edible coating prevents weight loss. The weight loss boosts TSS% in fruits [54]. Figure 1 indicates an overall reduction in TA of all treatments, but a higher reduction was shown in the control. TA typically declines with storing and a higher TA decline reveals senescence [20]. Edible coatings reduce gas interchange and therefore the fruit inhalation rate [53]. Respiration is an enzymatic procedure and the enzymes engaged in breathing use organic acids as a substratum [55].

Data shows the incremental increase in IL% and MDA accumulation in peach tissue during storage period (36 days), depending on the storing duration and treatments. MDA is usually used as an indicator of cell membrane degradation or oxidative injury since it is the result of lipid peroxidation [56]. Raised reactive oxygen species (ROS) accumulation caused more lipid peroxidation and increasing IL%, destroyed membranes, and reduced storing ability [57]. AVG 30% + CH 1.5% treated peaches displayed a decrease in IL% and MDA levels. AVG were matching with CH to prevention gas exchange, fruit respiration rate, and ethylene production [20,53].

Our results indicated that the edible coating combination of AVG 30% + CH 1.5% had significantly generated the DPPH radical scavenging effects and minimized generation rates of $H_2O_2$ and $O_2\bullet-$ in peach fruits. As earlier stated, in tomatoes and fresh-cut pears, we confirmed the findings of the existing study [58,59]. The results are in agreement with [60–62] who reported that combination of AVG+ CH might quench the creation of $H_2O_2$ and $O_2\bullet-$ by enhancing the antioxidant enzymes activities. Additionally, antioxidant activity varies on the fruit physiological position and declined during senescence [63,64]. Peach senescence was delayed by an edible coating combination. A positive suggestion of the DPPH radical-scavenging effect and post-harvest maturing procedure was conveyed [65].

CAT has been recognized as an antioxidant enzyme that shows a critical part in the oxidation-impervious effects of plants. Reactive oxygen species (ROS) are shaped in cells in abiotic and biotic pressure circumstances [66]. Numerous enzymes, namely SOD, CAT, and POD, are involved in the lessening of the ROS, especially CAT [67]. Additionally, POD partakes in the quick transformation of ($O_2\bullet-$) into $H_2O_2$ to conserve the cell from oxidative

pressure. In this context, it has been exposed that the utilization of AVG and CH on fruits as edible coating materials improves the antioxidant features in the fruits [60,61,68–71]. Polyphenols are measured to be key subordinate metabolites that have a strong influence on the quality (e.g., taste, color, and acidity) of fruits and have antimicrobial and antioxidant features [72]. It has already been reported that the phenol content increases in plants treated with CH [73]. Accordingly, [51] detected a rise in the phenol content of pumpkin seeds preserved with CH. Formerly, [74] reported higher phenols in table grapes covered with AVG. Coatings cause slight stress on fruits and complex coatings can upsurge the stress strength [51]. Therefore, adding AVG with a CH coating could synergistically maintain all of the phenol content in the peach fruit. In addition, phenols are antimicrobial components that are utilized by plants as a primary protection reaction versus incursive microorganisms. The gathering of phenols in plants rises with occurrence by pathogens. Accordingly, it has a positive experiential association between TPC content and antioxidant effects [75].

## 5. Conclusions

Both the Aloe vera-CH edible coating combination AVG 30% + CH 1.5% have excessive possibilities for prolonging the cold storing period of peach fruit (*Prunus persica* (L.) Metghamer, Sultany). AVG, in combination with CH coating, successfully retained quality features such as firmness, weight loss, skin color, TA, and TSS of peach fruit during cold storing. In addition, results show that AVG in a CH coating maintains the peach fruit's high TPC, antioxidant capacity, and quenches the generation of $H_2O_2$ and $O_2\bullet-$ during storing. This proposes that the AVG-CH coating effectively controls and minimizes ion leakage and malondialdehyde. Hence, it is decided that the combination of AVG and CH can be utilized to enhance storability and extend the cold storing period of peach fruit. Otherwise, the effects of other mixtures mediated between the effect of the best treatment and the control treatment, as they caused a higher improvement than the control, but less than the recommended treatment (AVG 30% + CH 1.5%). The current results indicate that the use of edible mixtures in fruit packaging for storage and prolongation of the marketing period still needs further research study to determine the proportions of mixtures that have the best effect and the least economic, while clarifying the physiological roles of the materials used.

**Author Contributions:** M.S.A. and M.S.G. perceived the idea and designed experiments. M.S.A. performed statistical analysis and design charts. M.S.A. and M.S.G. conducted the field experiments. M.S.A. and M.S.G. collected the data. M.S.A., S.F.E.-G., R.S., H.A., A.A. (Amal Alyamani) and A.A. (A. Almasoudi) analyzed the data and wrote first draft of manuscript. M.S.A., M.S.G., S.F.E.-G., R.S., A.A. (Amal Alyamani) and R.S. reviewed and prepared final draft of manuscript. All authors have read and agreed to the published version of the manuscript.

**Funding:** This research received no external funding.

**Institutional Review Board Statement:** Not applicable.

**Informed Consent Statement:** Not applicable.

**Data Availability Statement:** Available upon request from the corresponding author.

**Acknowledgments:** Taif University Researchers Supporting Project Number (TURSP-2020/140), Taif University, Taif, Saudi Arabia.

**Conflicts of Interest:** The authors declared no conflict of interest.

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
