# Peer review of "Synergistic Effect of Dipping in Aloe Vera Gel and Mixing with Chitosan or Calcium Chloride on the Activities of Antioxidant Enzymes and Cold Storage Potential of Peach (Prunus persica L.) Fruits"

_coatings, doi:10.3390/coatings12040498_

Round 1

Reviewer 1 Report

In the present study, the effect of Aloe vera gel, chitosan and CaCL2 on physical and chemical properties of peach fruit was evaluated.

The authors should not use very long sentences e.g. Line 21-25, 25-28, 101-105 etc.

The aim of the study should be clearly presented (Lines 21-25).

Line 27: what “extra significant” means?

Lines 87-89: present the aim of the study and the tested hypothesis in more detail.

Provide the ANOVA table which shows the p values for each factor and the interactions (storage x coating).

It is not clear if different fruit batches were used for weight loss and the chemical parameters? Was weight loss recorded for the same fruit throughout the storage period? And if yes, where there any other fruit used for the chemical and physical parameters analyses?

Correct the vertical axis of Figure 1 to “Soluble solids content”.

Check the letter of statistics in all the Figures. It is better to replace Figures with Tables.

Lines 316-318: weight loss is the result of transpiration and respiration. Therefore, the observed weight reduction should be allocated to both factors.

Lines 349-351: the increase of TSS should be also attributed to concentration effects due to water losses.

Author Response

reviewer # 1

Dear prof.

On my own behalf and on behalf of the co-authors, I thank you for your effort in reviewing the manuscript and I would like to inform you that all of your comments are useful and have led to the improvement of the research and made it rise to international publication.

Thank you very much

comment 1: The authors should not use very long sentences e.g. Line 21-25, 25-28, 101-105 etc.

response: These sentences are rewritten shortly

comment 2: The aim of the study should be clearly presented (Lines 21-25).

response: it was corrected and rewritten

comment 3: Line 27: what “extra significant” means?

response: Word  “extra" deleted

comment 4: Lines 87-89: present the aim of the study and the tested hypothesis in more detail.

response: it was corrected and rewritten  in more details

comment 5: Provide the ANOVA table which shows the p values for each factor and the interactions (storage x coating).

response: p-value has been added at the bottom of Table1, as well as at the bottom of each Figure

comment 5: It is not clear if different fruit batches were used for weight loss and the chemical parameters? Was weight loss recorded for the same fruit throughout the storage period? And if yes, where there any other fruit used for the chemical and physical parameters analyses?

response: This comment was explained in the Materials and  Methods  section on lines 112-115:

(The chosen peach fruits were divided into five groups each group consisting of 90 fruits with three replicates (30 fruits/replicate),each replicate was divided into two main batches. The first one ( 15 fruits ) was used for physical and chemical measurements while, the second batch was used to estimate weight loss.)

comment 6: Correct the vertical axis of Figure 1 to “Soluble solids content”.

response: It was corrected

comment 7: Check the letter of statistics in all the Figures. It is better to replace Figures with Tables

response: All letters in all figures were checked, Please suffice the first table and keep the rest of the charts

comment 8: Lines 316-318: weight loss is the result of transpiration and respiration. Therefore, the observed weight reduction should be allocated to both factors.

response: Edited and added to the line 318

comment 9: Lines 349-351: the increase of TSS should be also attributed to concentration effects due to water losses.

response: Yes, it is true, and thank you about this comment

Reviewer 2 Report

Comments:

This manuscript “Synergistic Effect of Dipping in Aloe Vera Gel and Mixing with Chitosan or Calcium Chloride on the Activities of Antioxidant Enzymes and Cold Storage Potential of Peach (Prunus persica L.) Fruits” aims to study the synergistic influence of coating with Aloe vera gel (AVG) at 15% or 30 % mixed with chitosan (CH) or calcium chloride (CaCl2) on physical and chemical features of peach fruits Prunus persica (L.) Metghamer Sultany.

The topic is not new and recent articles have been published:

  1. Van T.B. Nguyen, Duyen H.H. Nguyen, Ha V.H. Nguyen, Combination effects of calcium chloride and nano-chitosan on the postharvest quality of strawberry (Fragaria x ananassa Duch.), Postharvest Biology and Technology, Volume 162, 2020, https://doi.org/10.1016/j.postharvbio.2019.111103.
  2. Kou, XH., Guo, Wl., Guo, Rz. et al. Effects of Chitosan, Calcium Chloride, and Pullulan Coating Treatments on Antioxidant Activity in Pear cv. “Huang guan” During Storage. Food Bioprocess Technol 7, 671–681 (2014). https://doi.org/10.1007/s11947-013-1085-9

The topic studied is important and the article contributes to positive and interesting new data on the subject in question. The introduction summarizes the previous works. Proper experiments were conducted, and the data obtained is interesting, however, I think some points must be improved:

  1. In the text, please wright the species names in italic.
  2. In the figures, the alphabetical letters that indicate the significance should be in black because they are not easy to see.
  3. It is not mandatory, but, the article would be better if a sensory tasting panel was used to describe the possible sensory changes between the peaches coated and uncoated.

So, I think the article may be published after a minor revision.

Author Response

reviewer # 2

Dear prof.

On my own behalf and on behalf of the co-authors, I thank you for your effort in reviewing the manuscript and I would like to inform you that all of your comments are useful and have led to the improvement of the research and made it rise to international publication.

Thank you very much

comment 1: In the text, please write the species names in italic

response: All species names corrected and write  in italic

comment 2: In the figures, the alphabetical letters that indicate the significance should be in black because they are not easy to see.

response: Alphabetical letters in all figures changed to black colure

comment 3: It is not mandatory, but, the article would be better if a sensory tasting panel was used to describe the possible sensory changes between the peaches coated and uncoated.

response: I will consider this comment in my future research

Reviewer 3 Report

Comments on the Manuscript “Synergistic Effect of Dipping in Aloe Vera Gel and Mixing with Chitosan or Calcium Chloride on the Activities of Antioxidant Enzymes and Cold Storage Potential of Peach (Prunus persica L.) Fruits” referenced by coatings-1644710

This paper reports on the physical and chemical features of peach fruits Prunus persica (L.) coated with edible films based on Aloe vera gel mixed with chitosan or calcium chloride as safe platforms to prolong the storage and marketing period of peach fruits. Indeed, the development of safe and efficient natural coating films has been recognized as an emerging tool in the recent years because of ubiquitous and crucial role in minimizing fruit maturation and postharvest deterioration. I consider that the manuscript is suitable for publication in Coatings journal after Minor Revision. This paper has some issues, as it is presented in detail below:

  1. It is not clear the role of CaCl2 in the preparation of the edible films.
  2. How was optimized the concentration of Aloe vera gel, Chitosan and CaCl2. Why a concentration of 15% and 30% of AVG was selected? The same applies for CH (1.5%) and CaCl2 (3%). Please clarify this aspect in the section Materials and Methods.
  3. What is the thickness of the edible film?
  4. The results indicated the edible coating combination AVG 30% + CH 1.5% with the best performance. What about the other mixtures involved in this work. Please discuss also the effect of CaCl2 addition. It is not clear why CaCl2 has been added.
  5. The resolution of the Figures is too low. Please include captions and legends for all data included in the Figures. Not clear, in the Figures the meaning of “a,b,c..” for each data point plotted.
  6. Conclusions ought to have caveats, and information concerning gaps that prevail still. Also some lines about avenues created could help.

Author Response

reviewer # 3

Dear prof.

On my own behalf and on behalf of the co-authors, I thank you for your effort in reviewing the manuscript and I would like to inform you that all of your comments are useful and have led to the improvement of the research and made it rise to international publication.

Thank you very much

comment 1: It is not clear the role of CaCl2 in the preparation of the edible films.

response: Based on the information previously available in the introduction in lines 85-88, AVG extract has a vital role in reducing dehydration and maintaining the quality of fruits during post-harvest storing. Moreover, calcium has an effective physiological role in the formation of cell walls and regulation of metabolic activities during storage, so the current research aims to combine the two biological roles of both AVG and calcium

comment 2: How was optimized the concentration of Aloe vera gel, Chitosan and CaCl2. Why a concentration of 15% and 30% of AVG was selected? The same applies for CH (1.5%) and CaCl2 (3%). Please clarify this aspect in the section Materials and Methods.

response: The choice of concentrations of 15% and 30% of aloe vera is due to the Previous researches used high concentrations of AVG ranging from 50% to 100%, we applied the lower concentrations (15 -  30%) to reduce economic costs. As for the use of a concentration of 1.5 % of chitosan and 3% of calcium chloride, this is due to previous research, which stated that these two concentrations achieved the best effects on the chemical and physical properties of fruits, as well as contributed to prolonging the storage period and the trading period of fruits in the market.in the section Materials and Methods, we clarified this aspect.(line 123-125)

comment 3: What is the thickness of the edible film?

response: The thickness of the edible film was in general less than 0.3mm according Tharantharn (2003). (line 141)

comment 4: The results indicated the edible coating combination AVG 30% + CH 1.5% with the best performance. What about the other mixtures involved in this work. Please discuss also the effect of CaCl2 addition. It is not clear why CaCl2 has been added.

response:

-The effects of other mixtures mediated between the effect of the best treatment and the control treatment, as they caused a higher improvement than the control, but less than the recommended treatment.( added to  conclusions)

- Calcium chloride was added to the effective role in preserving the cell walls from deterioration as mentioned previously, and it was used in this research to study the extent of its effectiveness when mixed with Aloe Vera and chitosan, as it was not used before in this mixture.

comment 5: The resolution of the Figures is too low. Please include captions and legends for all data included in the Figures. Not clear, in the Figures the meaning of “a,b,c..” for each data point plotted.

response: The resolution of the Figures has been modified. As for the meaning of a-b-c, it shows the arrangement of the Figures and captions and legends for all data included in the Figures

comment 6: Conclusions ought to have caveats, and information concerning gaps that prevail still. Also some lines about avenues created could help.

response: This paragraph has been added to  conclusions:

The current results indicate that the use of edible mixtures in fruit packaging for storage and prolongation of the marketing period still needs further research study to determine the proportions of mixtures that have the best effect and the least economic, while clarifying the physiological roles of the materials used.

Round 2

Reviewer 1 Report

The authors revised the text according to the raised comments. Therefore, I suggest the acceptance of the manuscript in its present form.